# A Faster and More Accurate Iterative Threshold Algorithm for Signal Reconstruction in Compressed Sensing

**DOI:** 10.3390/s22114218

**Published:** 2022-06-01

**Authors:** Jianxiang Wei, Shumin Mao, Jiming Dai, Ziren Wang, Weidong Huang, Yonghong Yu

**Affiliations:** 1School of Management, Nanjing University of Posts and Telecommunications, Nanjing 210003, China; 2School of Internet of Things, Nanjing University of Posts and Telecommunications, Nanjing 210003, China; 1020072115@njupt.edu.cn (S.M.); b19060238@njupt.edu.cn (J.D.); 3School of Computer Science, Nanjing University of Posts and Telecommunications, Nanjing 210023, China; b20032110@njupt.edu.cn; 4Key Research Base of Philosophy and Social Sciences in Jiangsu-Information Industry Integration Innovation and Emergency Management Research Center, Nanjing 210003, China; huangwd@njupt.edu.cn; 5College of Tongda, Nanjing University of Posts and Telecommunications, Nanjing 210003, China; yuyh@njupt.edu.cn

**Keywords:** FISTA, compressed sensing, threshold function, restart adjustment mechanism, parameter adjustment, signal reconstruction

## Abstract

Fast iterative soft threshold algorithm (FISTA) is one of the algorithms for the reconstruction part of compressed sensing (CS). However, FISTA cannot meet the increasing demands for accuracy and efficiency in the signal reconstruction. Thus, an improved algorithm (FIPITA, fast iterative parametric improved threshold algorithm) based on mended threshold function, restart adjustment mechanism and parameter adjustment is proposed. The three parameters used to generate the gradient in the FISTA are carefully selected by assessing the impact of them on the performance of the algorithm. The developed threshold function is used to replace the soft threshold function to reduce the reconstruction error and a restart mechanism is added at the end of each iteration to speed up the algorithm. The simulation experiment is carried out on one-dimensional signal and the FISTA, RadaFISTA and RestartFISTA are used as the comparison objects, with the result that in one case, for example, the residual rate of FIPITA is about 6.35% lower than those three and the number of iterations required to achieve the minimum error is also about 102 less than that of FISTA.

## 1. Introduction

At the turn of the twentieth century, the compressed sensing (CS) theory emerged in the field of signal processing [1,2]. According to studies, the CS is particularly well suited to wireless sensor networks (WSN) and has a promising future in wireless data communication [3,4,5,6,7].

To reconstruct the original signal with high precision, we require a faster convergence speed with less sampling information in the reconstruction algorithm [8] of CS. Iterative threshold algorithm (ITA), a convex optimization algorithm, converts the reconstruction problem into a convex optimization problem that can be solved by linear programing. To begin, the “threshold” in the algorithm refers to the hard [9] and soft threshold functions [10]. Second, the lasso optimization problem was studied with a gradient descent algorithm [11] and an iterative threshold function. The proximal gradient algorithm (PGA) [12] was also created by the researchers to solve the lasso problem [13]. Overall, the iterative soft threshold algorithm can be regarded as a combination of the PGA and the soft threshold function. The “soft threshold” is utilized as the gradient of the objective function and the “gradient descent” is employed to obtain the best value. Researchers have raised numerous improved FISTAs, such as AFISTA, [14] which fastens the FISTA by a continuation strategy, and S-FISTA [15], which uses a scaling technique for gradient proximal step. EFISTA [16], monotonic FISTA [17], restart FISTA [18,19] and backtracking strategy [20] are also available. There is no doubt that those algorithms can play a larger role in a variety of fields [21,22,23].

This study will put forward a new improved FISTA. We will firstly suggest a better threshold function and demonstrate theoretically that it can overcome the discontinuity and constant deviation of the classical threshold function. Simultaneously, the study will integrate three parameters and the restart judgment mechanism. In this way, an improved iterative threshold algorithm with a higher convergence rate and better reconstruction performance would be formed.

## 2. Related Work

### 2.1. Basic Theory

In the sensor network structure module, compressed sensing is essential. Figure 1 depicts the basic steps of compressed sensing.

In the reconstruction model, as shown in the above graph, the observation vector $y\in R^{M}$ can be expressed as:(1)$$y=\phi x=\phi \psi^{-1}S=\Theta S,$$

To solve such a linear inverse problem, the least square method is usually used and the form is as follows:(2)$${\overset{\wedge }{x}}_{LS}=\underset{x}{\arg\min}||y-\phi x|{|}_{2}^{2},$$

To attain decent results, such a morbid linear formula must avoid significant estimation variance caused by unbiased estimation and the Tikhonov regularization approach is employed. Simultaneously, Equation (3) is formed, which is the classical lasso problem [13] due to the benefit that the regular term of the L1 norm can produce sparse solutions,
(3)$$\underset{x}{\min}\frac{1}{2}||y-\phi x|{|}_{2}^{2}+\lambda ||x|{|}_{1}^{2},$$

Among this expression, λ>0 is the regularization parameter.

### 2.2. Fast Iterative Soft Threshold Function Reconstruction Algorithm Based on Proximal Gradient Descent

It is clear that the basic optimization problem described in Equation (3) can be solved by the proximal gradient descent method.

After simplification and optimization, it can be calculated like this:(4)$$\begin{array}{l} x^{k}=prox_{t,\lambda ||\cdot |{|}_{1}}(x^{k-1}-t\nabla g(x^{k-1}))\\ =S_{\lambda t}(x^{k-1}-t\phi^{T}(\phi x^{k-1}-y)) \end{array}$$
where $S_{\lambda t}()$ is the soft threshold operator, g(x) is the first half of Equation (3) and is described as $g(x)=\frac{1}{2}||y-\phi x|{|}_{2}^{2}$. In addition, it has $\nabla g(x)=\nabla \frac{1}{2}||y-\phi x|{|}_{2}^{2}=\phi^{T}(y-\phi x)$ and t is achieved with the Lipchitz continuity in $R^{n}$ of ∇g(x), which means $\forall x_{1},x_{2}\in R^{n},\exists L>0$ satisfies $\left|\nabla g(x_{1})-\nabla g(x_{2})|\leq L|x_{1}-x_{2}\right|$. Then, set $t=\frac{1}{L}$.

Nevertheless, according to the reference [24], for the objective Equation (3), the convergence rate of $f(x_{k})-f(x^{*})$ is $O(\frac{1}{k})$ and so is the time complexity of ISTA. Fast iterative soft threshold algorithm (FISTA) is created using Nesterov acceleration technology to speed up the convergence speed of the algorithm. The difference between FISTA and ISTA lies in the Nesterov accelerating process, which only requires a few additional steps and brings great elevation to the convergence speed of the algorithm. ISTA mainly relies on the value of approximate function $x_{k-1}$ from the previous iteration and the Function (4) is the only kernel of the algorithm. FISTA makes use of the Nesterov acceleration technology and it mainly consists of two parts. The first part is computing a new point along the direction of the previous two steps. In the second part, the cardinal point is processed by approximate gradient method. The core formula of the algorithm is shown in Formula (5).
(5)$$\{\begin{array}{c} x_{k+1}=S_{\lambda t}(y_{k}-t_{k}\nabla f(y_{k}))\\ \xi_{k}=\frac{1+\sqrt{1+4\xi_{k-1}^{2}}}{2}\\ \gamma_{k}=\frac{\xi_{k-1}-1}{\xi_{k}}\\ y_{k}=(1-\gamma_{k})x_{k}+\gamma_{k}x_{k-1} \end{array},$$

Therein, ξ and γ are two momenta, *k* represents the *k*-th iteration and we set $\;\xi_{1}=1$, $y_{1}=x_{0}\in R^{n}$.

## 3. Proposed Method

### 3.1. Parameter Variation Based on Nesterov

As seen in the previous section, there are the following calculation rules in the FISTA
(6)$$\xi_{k}=\frac{1+\sqrt{1+4\xi_{k-1}^{2}}}{2},\gamma_{k}=\frac{\xi_{k-1}-1}{\xi_{k}},$$

Obviously, each change of $x_{k}$ is related to the value of the $\gamma_{k}$ and the $\gamma_{k}$ is constantly affected by the $\xi_{k}$. Two constants “1” and one constant “4” are required for each update in the calculation rules for $\xi_{k}$. In this paper, the three fixed constants are replaced by three parameters *p*, *q* and *r* to study the impact of those three parameters on the iteration process. Therefore, the calculation rules that integrate the three parameters *p*, *q* and *r* are modified as follows:(7)$$\xi_{k}=\frac{p+\sqrt{q+r\xi_{k-1}^{2}}}{2},\gamma_{k}=\frac{\xi_{k-1}-1}{\xi_{k}},$$

First, analyzing the convergence is necessary and here is the specific analysis:(8)$$\begin{array}{l} r\in (0,4):\xi_{k}\to \frac{2p+\Delta }{4-r}<+\infty ,\gamma_{k}\to \frac{2p+\Delta -(4-r)}{2p+\Delta }<1,\\ r=4:\xi_{k}\approx \frac{k+1}{2}p\to +\infty ,\gamma_{k}\to 1 \end{array}$$
where $\Delta \overset{def}{=}\sqrt{rp^{2}+(4-r)q}$. It indicates that only when the value of *r* is 4 can the $\gamma_{k}$ converge to 1 and the convergence rate of $f(x_{k})-f(x^{*})$ is $O(\frac{1}{k^{2}})$. It also has the advantage of proving the convergence in the iterative process thanks to this constant. So we could set *r* as 4 for the algorithm. The values of *p* and *q*, on the other hand, can have a difference on the convergence of the algorithm to a certain extent. Let $x^{*}$ be the optimal solution of the lasso minimum problem and lead $x^{*}$ and $x_{k}$ into Formula (2) to get the solution of their respective minimum errors. According to relevant research, let $f(x)=\frac{1}{2}||y-\phi x|{|}_{2}^{2}+\lambda ||x|{|}_{1}^{2}$, then the convergence of $f(x_{k})-f(x^{*})$ is consistent with that of $\left\{x_{k}\right\}$.

The data in the Figure 2 are made up of random signals and the findings are achieved through several experiments. It can be seen clearly that FISTA has an oscillation problem, which has a detrimental impact on the number of iterations, resulting in time waste and a drop in reconstruction efficiency. 

Relevant studies have proposed that some judgment restart mechanisms be implemented at the end of each iteration process to alleviate these problems. According to the scheme presented in reference [18], the restart mechanism can be incorporated and forms RadaFISTA, which means that FISTA is regarded as a generalized gradient scheme and Formula (9) is a generalized gradient step. By such mechanism, the new algorithm is able to achieve an almost monotonic convergence in terms of $f(x_{k})-f(x^{*})$ and can get a significantly faster speed.
(9)$$x_{k+1}=S_{\lambda t}(y_{k}-t_{k}\nabla f(y_{k})),$$

The gradient restart scheme is equivalent to making judgement on ${\left(y_{k}-x_{k+1}\right)}^{T}(x_{k+1}-x_{k})>0$ before each iteration. At the same time, as resetting $y_{k}$ in each iteration, it can never be neglected to regulate the value of r and add a factor between 0 and 1 to adjust and control the range of r value within 4.

When determining the value of these two parameters, *p* and *q*, we first fix that they must both be positive. Then, several data pairs composed of the values of *p* and *q* are formed and make up a data set. To achieve the suitable values for the parameters *p* and *q*, we use hundreds of groups of random signals as numerical instances and conduct 100 experiments on each group of signals. As shown in the Figure 3 and Figure 4, the data in those figures are made up of random signals and the findings are achieved through those experiments.

On the premise that the value of *p* is fixed at 1, take the *q* values of ½, 1, 2 and 10, respectively, for relevant experiments. As shown in Figure 3, the value of *q* has little effect on the results in those multiple experiments. Consequently, in order to facilitate the implementation of the following experiments, we set the value of *q* as 1 in the algorithm.

The choice of *p* value, on the other hand, has an impact on the convergence. Keep the *q* value constant and take *p* values of 1/20, 1, 2 and 4, respectively, for the experiment. The results are exhibited in Figure 4.

Through repeated tests, it is seen that when the value of *p* is taken as 2, relatively still less data residuals can be obtained in fewer iteration times. The effect of taking *p* value around 2 cannot be achieved by other values. Therefore, in order to bring convenience for the following experiments, the value of *p* is set to 2 in the improved algorithm given in this paper.

Apparently, the parameters *p* and *q* appear to be able to control the reconstruction process after being added into the algorithm. In comparison to FISTA, this algorithm proposed in this paper can more easily control the convergence rate and the reconstruction speed by adjusting the parameters *p* and *q*, making it more versatile and adaptable to signals with different characteristics.

### 3.2. Improved Threshold Function

For the sake of seeking the “gradient” of the objective function containing the L1 norm, a soft threshold function is adopted to calculate the optimal solution. Evidently, the soft threshold function is widely used and has shown to be extremely reliable. However, both the soft threshold function and the hard threshold function have flaws. Consequently, this paper proposes a new threshold function that combines the features of soft and hard threshold functions and applies them throughout the reconstruction algorithm.
(10)$$\overset{\wedge }{x_{i}}=\{\begin{array}{l} 0,\;\;\;\;\;\;|x_{i}|<\lambda \\ x_{i}(1-{\left(\frac{\lambda }{\left|x_{i}\right|}\right)}^{n}),|x_{i}|\geq \lambda \end{array},$$

Therein, $x_{i}$ is the processed values of $y_{k}$ and n is a variable quantity that gives the enhanced threshold function a lot of flexibility. Furthermore, the threshold function proposed in this paper also has the advantage of a concise expression, which avoids the problems of multiple parameters and significant inconvenience that many threshold functions have to some extent. Additionally, additional advantages are as follows. In terms of continuity, it can be seen from Formula (10) that the function is continuous at ±λ, which allows it to avoid the defects of the hard threshold function and to smooth out the signal. Additionally, when the coefficient $x_{i}$ tends to infinity, the deviation of the soft threshold function will lead to distortion. However, in this improved function, it can be seen that $\underset{x_{i}\to +\infty }{\lim}\overset{\wedge }{x_{i}}-x_{i}=0$, $\underset{x_{i}\to -\infty }{\lim}\overset{\wedge }{x_{i}}-x_{i}=0$. Hence, the improved threshold function described in this study can partially overcome the constant deviation of the soft threshold function and it is also verified that the deviation will have less and less influence on the updated threshold function.

Figure 5 illustrates that along with the increase of $x_{i}$, applying the improved threshold function described in this paper results in greater sparse constraints. The new function produces smaller deviations for large coefficients $x_{i}$ than the soft threshold function. By keeping large coefficients $x_{i}$, we could overcome the problem of easy distortion of the soft threshold function and by shrinking the intermediate coefficients $x_{i}$, we could reduce the discontinuity of the hard threshold function. When the value of *n* is 1, the function is a soft threshold function. When the value of *n* approaches infinity, the value of $x_{i}$ approaches the value of the hard threshold function. To some extent, the better function compensates for the shortcomings of constant deviation. Furthermore, after processing, the estimated coefficient $\overset{\wedge }{x_{i}}$ obtained can approach the real value more quickly. Meanwhile, it also makes up for the discontinuity of the hard threshold function at the threshold ±λ.

As a result, an improved threshold function is constructed for the lasso problem in the compressed sensing reconstruction algorithm. In the reconstruction algorithm, the iterative threshold algorithm attempts the solution to the lasso problem indicated in Formula (2). The soft threshold function, as seen in Section 2.2, is frequently utilized to solve such reconstruction problems. However, the improved function is used to displace the soft threshold function in this paper and is integrated into the FISTA core formula.
(11)$$x_{k+1}=T_{\lambda n}(y_{k}-t_{k}\nabla f(y_{k})),$$

In this formula, $T_{\lambda n}$ denotes the improved threshold function, λ is the threshold, n is the function variable and $t_{k}$ is the step size. These new penalty functions can be designed flexibly by changing the value of n in order to obtain a better reconstruction effect.

### 3.3. Improved Fast Iterative Threshold Algorithm

Aiming at improving the convergence speed and the reconstruction performance of FISTA, this paper puts forward a new algorithm named fast iterative parametric improved threshold algorithm (FIPITA), which substitutes the improved threshold function for the soft threshold function, uses a restart and self-adaptive adjustment mechanism to alleviate the oscillation problem in the reconstruction process and improve the convergence efficiency and integrates three parameters *p, q* and *r* to replace three previous constants. The final reconstructed signal can be obtained by combing the three elements above into the FIPITA.

At last, in order to facilitate the description and understanding of the execution steps of the above algorithm, the solving process is organized in the form of the following Algorithm 1.
**Algorithm 1** Fast iterative parametric improved threshold algorithm(FIPITA)**Input :**Lipschitz constant: *L = L(f)(L(f)*-A Lipschitz constant of ∇f)Initial value: *x*_0_**Output:**  Optimal value f(x) with *x.*1: **Begin**2: Initialize momentum1 ξ, momentum2 γ,3: Set  ξ1=1; y1=x0∈Rn; 4: **For** *k* = 1, 2, 3 … **COMPUTE**5:  xk+1=Tλn(yk−tk∇f(yk));6:  ξk=p+q+rξk−1227:  γk=ξk−1−1ξk8:  yk=(1−γk)xk+γkxk−19: **Restart if**
(yk−xk+1)T(xk+1−xk)>010: **Let**
r=ζr11:  yk=xk12:    **if**
r<3.9913:       **Reset** ξk14: **End For**15: Obtain the optimal LASSO answer *f(x)* and its corresponding *x*;16: **End**

## 4. Results and Discussions

Simulation experiments used to test and validate the fast iterative parametric improved threshold algorithm (FIPITA) are divided into several sections. These sections include comparing the signal before and after reconstruction to assess the reconstruction effectiveness, adopting the residual rate to assess the change in reconstruction accuracy with sparsity, adopting the required maximum number of iterations to assess the change in algorithm efficiency with sparsity and adopting the residual rate to assess the variation in reconstruction accuracy with the observed value. FISTA, the most classic and widely used algorithm, is added in the experiment as a comparable algorithm to further demonstrate the performance of the algorithm. Meanwhile, RestartFISTA [25] and RadaFISTA [18] are also added.

### 4.1. One Dimensional Signal Reconstruction Simulation Test

To demonstrate the property of the FIPITA, we take a Gaussian random signal x with length n of 256, observation value m of 128 and sparsity K of 10. The Gaussian random matrix is selected as the observation matrix of this simulation experiment. When the error of two adjacent iterations, $res=x_{k}-x_{k-1}$, is less than 10^−16^ or the value of $y-\phi x_{k}$ is less than 10^−6^, the algorithm has to call off the iteration.

From Figure 6, it is be indicated that the signal can be rebuilt accurately by the mended algorithm.

### 4.2. Performance of the Algorithm under Different Sparsity

The residual rate is a key verification index for evaluating the algorithm performance and it is defined as follows:(12)$$residualrate=\frac{\left||\overset{\wedge }{x}-x|\right|}{\left||x|\right|},$$

Among Formula (12), $\overset{\wedge }{x}$ stands for the final reconstruction result, *x* is the original signal and the ratio of the norm of $\overset{\wedge }{x}-x$ to the norm of the original signal *x* is deeded as the index of signal reconstruction quality; the smaller the index the higher the signal reconstruction accuracy.

FISTA, RadaFISTA and RestartFISTA are drawn into the experiment to compare with the FIPITA. Gaussian and Hadamard matrices are selected as observation matrices for measurement experiments at the same time. The type, length of the signal and the observation value are identical to those stated in the previous section. The sparsity K is set between 21 and 70 and the step size is 1. The same group of signals is reconstructed by those four algorithms.

According to Figure 7a,b, the residual rate of the modified iterative threshold algorithm is the lowest and does not change with sparsity or observation matrix. Overall, the residual rates of the four algorithms are steadily increasing in tandem with the increase in sparsity, which is in keeping with the regular pattern of the algorithm. Moreover, as shown in the picture above, the residual rate of the FISTA, RadaFISTA and RestartFISTA are pretty close, that is, the reconstruction effects of these three algorithms are similar. Taking the Gaussian matrix as the observation matrix and the interval of sparsity between 21 and 70 as an example, the residual rate of the FIPITA proposed in this paper is approximately 6.35% lower than the other three algorithms. According to the definition of residual rate, the lower the residual rate the higher the reconstruction accuracy, which suggests the reconstruction accuracy of the FIPITA is supreme.

### 4.3. Performance of the Algorithm under Different Observation Numbers

The original signal is a Gaussian random signal with the same signal type and length as the ones above. The observation value m is set to range between 75 and 125, the step size is 1 and the sparsity K is set to 50. The FISTA, RadaFISTA and RestartFISTA are still used to reconstruct the same group of signals with FIPITA and the Gaussian matrix and Hadamard matrices are used as the observation matrices, respectively. The measured residual rate is shown in the graphs below.

Although the observation value is varied, the residual rate of the FIPITA is the minimum among the four algorithms under different measurement matrices, as is clearly shown in Figure 8a,b, and the residual rate of those three algorithms used for comparison is close. Taking the Hadamard matrix as the observation matrix and the interval of observation value between 75 and 125 as an example, the residual rate of FIPITA in this study is around 4.99% lower than the other three algorithms. In other words, the FIPITA has the maximum reconstruction accuracy, whereas the other three algorithms have similar reconstruction accuracy.

### 4.4. Algorithm Efficiency Comparison

In this paper, the maximum number of iterations required by the algorithm is adopted as an index to measure the efficiency of the algorithm. The maximum number of iterations indicates the number of final iterations required when $res=x_{k}-x_{k-1}$ is less than 10^−16^ and the iteration is ceased. The efficiency of the algorithm improves as the number of iterations decreases, which means the reconstruction will be faster. Figure 9a,b shows the number of iterations for varied sparsity and observation matrices.

In light of Figure 9a,b, it is explicit that the number of iterations required by FISTA is the highest for various sparsity, implying that the algorithm requires more time cost and is inefficient. The FIPITA appears to be better than FISTA in terms of the number of iterations, but at the same time as sparsity increases the number of iterations required by the FIPITA is sometimes higher than RadaFISTA and RestartFISTA. For this reason, this paper integrates data from one particular experiment on the Hadamard matrix and displays them in the Table 1 below.

The statistics from Table 1 leads us to the conclusion that, while FIPITA will result in marginally higher iteration times than RadaFISTA and RestartFISTA, such iteration times are inherently unstable. When the sparsity is 30, for example, the number of its iterations is much less than the other three algorithms, including FISTA, and when the sparsity is 50, its iteration times are similar to RadaFISTA and RestartFISTA. At the same time, when it comes to residual rate, the FIPITA has a clear advantage, which can improve the accuracy of the compressed sensing reconstruction algorithm.

It is also important to combine the number of iterations with convergence. Several hundreds of experiments are done on the same group of signals to study the convergence.

The horizontal axis of Figure 10 denotes the number of iterations. We can see that the results of the figure are consistent with those analyzed above. The convergence rate and the iterations of FIPITA are similar to RadaFISTA and RestartFISTA and superior to FISTA. In addition, the convergence rate of $f(x_{k})-f(x^{*})$ is $O(\frac{1}{k^{2}})$, which is coincident with those three.

Generally speaking, while the number of iterations required by the suggested technique is significantly fewer than that of FISTA, it is slightly more than RadaFISTA and RestartFISTA. However, after thoroughly evaluating and judging the unstable nature of the number of iterations, lower residual rate and superior reconstruction performance of FIPITA in a comprehensive way, we can get a conclusion that the FIPITA not only has relatively rapid reconstruction speed but also can increase signal reconstruction accuracy.

## 5. Conclusions

The fast iterative parametric improved threshold algorithm (FIPITA) has combined the restart mechanism with the idea of backtracking and added three parameters so that it can converge faster than the FISTA. Furthermore, instead of using the soft threshold function to lower the residual rate of the algorithm, this paper uses the enhanced threshold function to improve the reconstruction accuracy. Through experiments, the effectiveness of the algorithm is tested and confirmed. The results reveal that this algorithm is not only better than several camparable algorithms in terms of reconstruction accuracy but also considerably superior to FISTA in terms of algorithm efficiency. Therefore, the next research project should focus on how to further improve the efficiency of the reconstruction algorithm while still ensuring reconstruction accuracy.

## Figures and Tables

**Figure 1 sensors-22-04218-f001:**
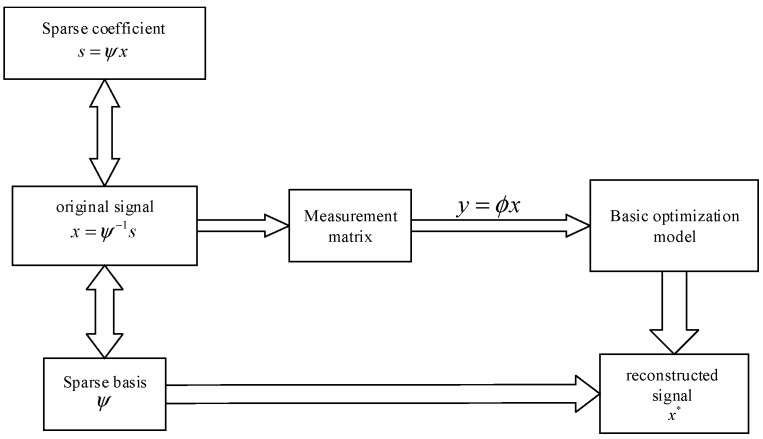
Basic framework of compressed sensing. (*x* denotes the original signal, and the *x** denotes the reconstructed signal).

**Figure 2 sensors-22-04218-f002:**
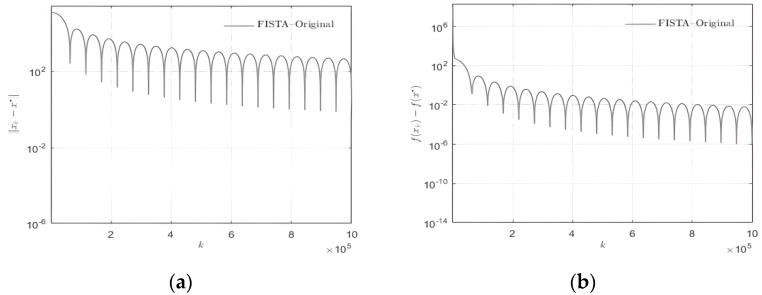
Iteration of conventional FISTA. (**a**) $\left||x_{k}-x^{*}|\right|$. It depicts the reconstruction error diagram of $x_{k}$ and $x^{*}$ in each iterative process under the traditional fast iterative soft threshold algorithm; (**b**) $f(x_{k})-f(x^{*})$ shows the difference between the value of $x_{k}$ obtained in each iteration and $x^{*}$ in solving the lasso problem shown in Formula (2).

**Figure 3 sensors-22-04218-f003:**
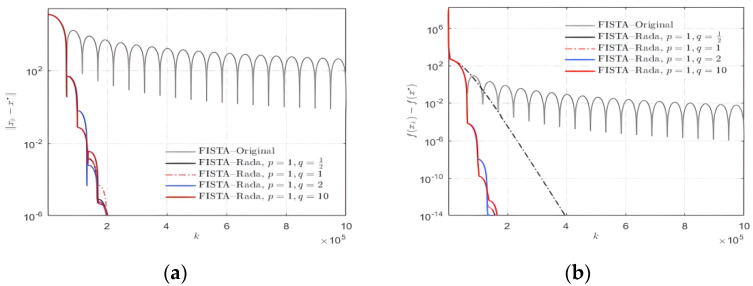
Selection of parameter *q* value for FISTA–Rada. (**a**) $\left||x_{k}-x^{*}|\right|$; (**b**) $f(x_{k})-f(x^{*})$.

**Figure 4 sensors-22-04218-f004:**
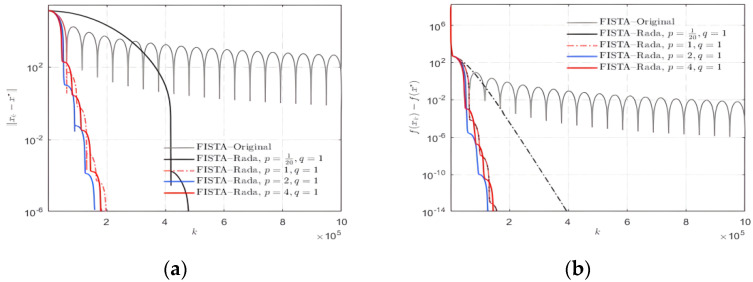
Selection of parameter *p* value for FISTA–Rada. (**a**) $\left||x_{k}-x^{*}|\right|$; (**b**) $f(x_{k})-f(x^{*})$.

**Figure 5 sensors-22-04218-f005:**
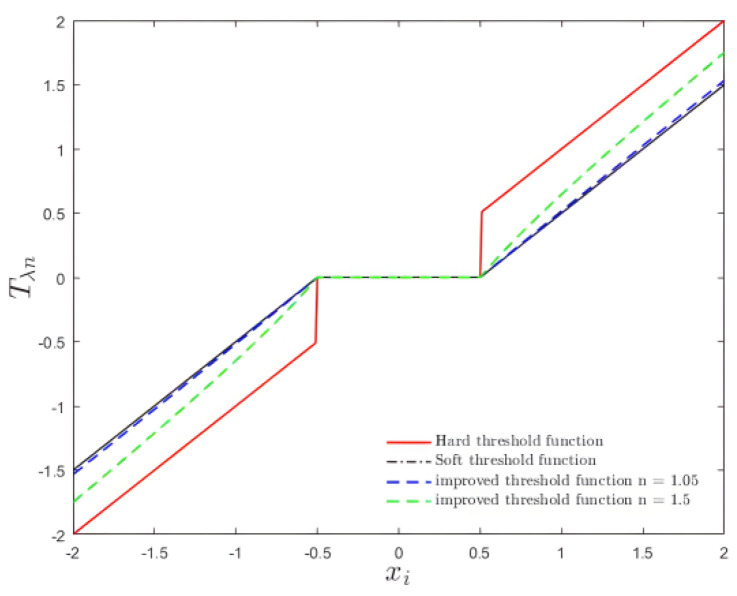
Threshold function diagram.

**Figure 6 sensors-22-04218-f006:**
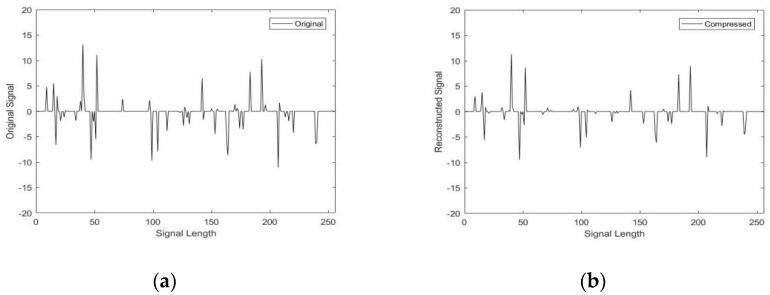
Signal reconstruction diagram. (**a**) Original signal; (**b**) the reconstructed signal.

**Figure 7 sensors-22-04218-f007:**
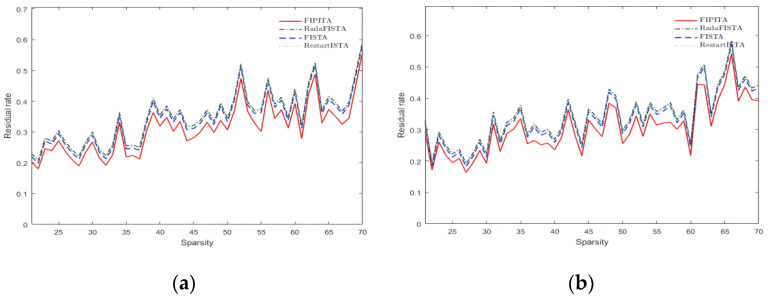
Signal residual rate under different sparsity. (**a**) Residual rate of Hadamard matrix; (**b**) residual rate of Gaussian matrix.

**Figure 8 sensors-22-04218-f008:**
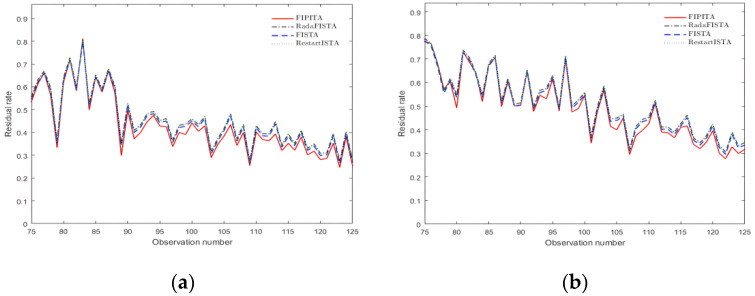
Signal residual rate under different observation numbers. (**a**) Residual rate of Hadamard matrix; (**b**) residual rate of Gaussian matrix.

**Figure 9 sensors-22-04218-f009:**
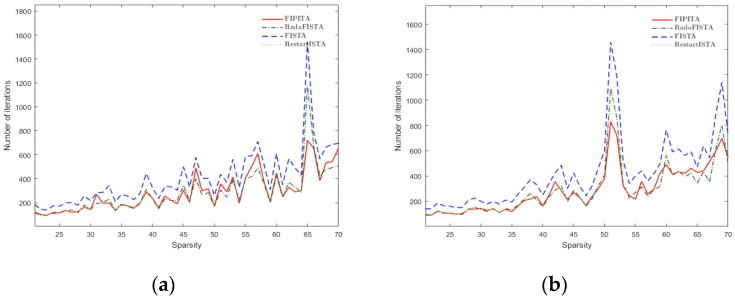
Required number of iterations under different sparsity. (**a**) Maximum number of iterations of Hadamard matrix; (**b**) maximum number of iterations of Gaussian matrix.

**Figure 10 sensors-22-04218-f010:**
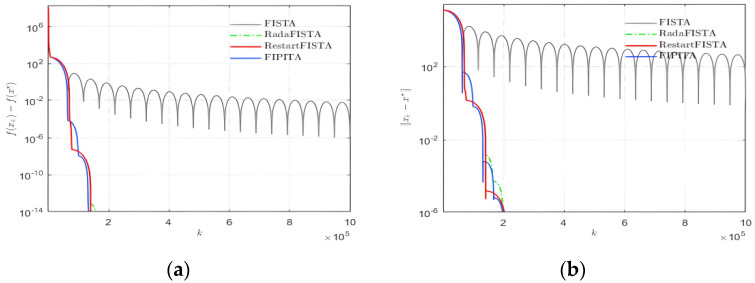
Iteration times and convergence. (**a**) $f(x_{k})-f(x^{*})$; (**b**) ||$x_{k}-x^{*}$||.

**Table 1 sensors-22-04218-t001:** Comparison of reconstruction residual rate and iteration numbers of four algorithms.

	Average Iteration Times under Different Sparsity	Sparsity (=30)		Sparsity (=45)		Sparsity (=50)	
		Number of Iterations	Residual Rate	Number of Iterations	Residual Rate	Number of Iterations	Residual Rate
FISTA	397.66	222	0.2861	346	0.2605	508	0.3609
RadaFISTA	273.82	150	0.2861	240	0.2602	357	0.3612
RestartFISTA	273.82	149	0.2860	237	0.2608	359	0.3610
FIPITA	295.92	145	0.2592	253	0.2423	358	0.3427

## Data Availability

The data of this study are available from the corresponding author upon request.

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
