# Peer review of "A Faster and More Accurate Iterative Threshold Algorithm for Signal Reconstruction in Compressed Sensing"

_sensors, 2022, doi:10.3390/s22114218_

Round 1

Reviewer 1 Report

General Comments:

The paper deals with compressed sensing in sensor networks and presents an improved version of the FISTA reconstruction algorithm which has better convergence and accuracy of the existing FISTA versions. The topic is important for sensor networks and the presented algorithm is interesting.

Specific Comments:

1. The Title, Keywords, and Abstract should refer to compressed sensing.

2. The Abstract should clearly define the shortcomings of the existing methods and how these shortcomings are dealt with. A quantitative comparison would be more effective.

3. Section 2: Please mention references for Sub-Section 2.2, e.g., clarify or reference the complexity of ISTA.

4. Section 3:

a) It is not clear how the suitable (optimal) values for the parameters p,q,r are selected.

b) The properties of the threshold function in Equation (9) should be fully analyzed, e.g., to clarify why the constant deviation of soft threshold function is considered a problem.

5. Section 4: The complexity of the proposed algorithm is not clear. Faster convergence could be observed versus the number of iterations, but that number does not give an idea about the actual complexity. Both time-complexity and hardware-complexity are important for sensor networks.

Language Usage:

The paper is in need for a moderate language revision. Just as examples:

  1. In the Abstract, the first paragraph is weakly written. The word “advanced” should be “proposed”.
  2. In the Abstract, “The parameters of the three constants” should be “The three parameters”.
  3. In the Introduction, “To achieve a high-precision original signal” should be “To reconstruct the original signal with high precision”.

… etc.

Round 2

Reviewer 1 Report

The Authors have successfully addressed all of the Reviewer’s comments. The current version is suitable for publication in MDPI Sensors.

One final comment: It would be better to replace the phrase “and Compressed Sensing” in the paper Title by the phrase “in Compressed Sensing”.